# Research on the Flexible Heating Model of an Air-Source Heat Pump System in Nursery Pig Houses

Hua Wang, Jijun Liu, Zhonghong Wu, Jia Liu, Lu Yi, Yixue Li, Siqi Li and Meizhi Wang *

College of Animal Science and Technology, China Agricultural University, Beijing 100193, China; wanghuatzlm@163.com (H.W.); liujijun@cau.edu.cn (J.L.); wuzhh@cau.edu.cn (Z.W.); liujiafg@163.com (J.L.); 18315154236@163.com (L.Y.); lyx200120@163.com (Y.L.); 15162358059@163.com (S.L.)
* Correspondence: meizhiwang@cau.edu.cn

**Abstract:** Maximizing the utilization of renewable energy for heating is crucial for reducing energy consumption in pig houses and enhancing energy efficiency. However, the mismatch between peak solar radiation and peak heat load demand in nursery pig houses results in energy waste. Therefore, we investigated a flexible air-source heat pump system (F-ASHP) based on the hourly-scale energy transfer of solar energy. A theoretical calculation model for F-ASHPs in pig houses in the heating areas of northern China has been established through on-site testing and Simulink. This study investigated the heat storage and release of four energy storage materials in pens and the variation in heat load in the house, validating the accuracy of the model. The results show that the F-ASHP can effectively match the peak solar heat and peak heat load in the house. Among the four energy storage materials in pens, the magnesium oxide heat storage brick material performed the best. During intermittent solar periods, it released 3319.20 kJ of heat, reducing the heat load in the pig house by 10.1% compared with that by the air-source heat pump (ASHP). This study provides a theoretical model for flexible heating calculations in pig houses in northern China and aims to serve as a valuable resource for selecting energy-storage pens.

**Keywords:** energy saving; F-ASHP model; nursery pig; storage pen; Simulink

## 1. Introduction

With the proposal of China's carbon peaking and carbon neutrality development goals, reducing energy consumption and increasing energy utilization efficiency in buildings have become increasingly important research topics across various fields [1–4]. Cold stress can significantly affect animal health and breeding, and an appropriate temperature is crucial for healthy nursery pig farming [5–8]. Nursery pigs require higher indoor temperatures, necessitating additional heating equipment in pig houses [9,10]. In recent years, the energy-saving and emission-reduction effects of pig houses have improved with the enhancement of the insulation performance of the enclosure structure and the application of clean energy [11,12]. However, in accordance with the "dual carbon" goal, building energy systems face higher development requirements to satisfy the need for further innovative low-carbon goals based on energy-saving measures. Therefore, researching the transition from energy-saving to low-carbon livestock houses, reducing heat load, improving heating efficiency in piglet houses, and promoting a flexible heating system that fully utilizes renewable energy to satisfy the fluctuating energy demand in livestock houses are significant for achieving low-carbon development in piglet houses and fostering a sustainable livestock industry.

Air-source heat pumps (ASHPs) are widely used for heating livestock houses, and their heating capacity is influenced by the external environmental temperature [13]. As the environmental temperature decreases, the evaporation temperature and pressure decrease, leading to an increase in the compressor pressure ratio and compressor power consumption and decreases in heating capacity and coefficient of performance (COP) [14,15].

Due to the intermittent and unstable nature of solar energy, the peak heating capacity of solar energy does not align with the peak heat load of livestock houses within a day [16]. In recent years, researchers have explored two main approaches to address these issues. The first approach involves combining solar energy with ASHP for heating purposes. Kim [17] designed a combined solar energy and ASHP system and conducted performance testing and optimization analysis, resulting in a 5.1% increase in the system's COP. Besagni [18] focused on optimizing the solar energy-coupled ASHP system by incorporating additional components, such as photovoltaic panels and inverters, into the collector, thereby creating a solar photovoltaic-thermal-coupled heat pump system. The results demonstrated a COP of up to 3.8, leading to improved thermal efficiency. Furthermore, Sezen [19] investigated the impact of ambient temperature on solar energy-coupled heat pumps and identified the optimal system COP for heating performance across different ambient temperature ranges.

The second approach involves the utilization of heat storage within the building envelope during peak solar periods and subsequent heat release during intermittent solar periods. Choi [20] conducted a study on a residential solar-air heating system that utilized foundation concrete as the heat storage material. The results indicated that the heat accumulated in the foundation concrete during the day could be effectively utilized during non-collection periods. By incorporating energy storage materials into building components, we can entirely exploit their heat storage and release characteristics, minimize heat waste during periods of non-utilization, and significantly enhance indoor thermal comfort.

Furthermore, Park [21] conducted experimental investigations on the thermal performance of phase-change bricks doped with phase-change materials in both summer and winter environments, comparing their effectiveness in controlling indoor temperatures with different roof materials. The results revealed that, during winter, the phase-change cold roof significantly improved the insulation performance of the roof structure, thereby reducing indoor heat loss. In another study, Ye [22] implemented the storage of cold energy within a building wall using a summer jet attachment. This approach enabled the utilization of stored cold energy at night, resulting in a reduction of the building's cooling load during summer daytime hours.

Heat pumps are highly efficient devices commonly used for cooling and heating purposes that can be integrated with renewable energy systems [23]. Simulink, an essential component of MATLAB, provides an integrated environment for modeling, simulating, and conducting comprehensive analyses of dynamic systems. Simulink has demonstrated excellent results in solving various time-varying system differential equations and can be employed for simulating the thermal environment within buildings [24,25].

The optimization of ASHP performance and building energy storage research is primarily based on residential and public buildings. The architectural structure of pig houses is different from that of residential buildings owing to the widespread use of special production processes, such as individual stalls, slatted floors, and exhaust fans. To date, research on storing renewable energy in energy storage pens through heating systems has not been conducted in pig houses for conservation. This study proposes a flexible air-source heat pump system (F-ASHP) model based on the hourly scale energy transfer of solar energy in a nursery pig house. Figure 1 shows a schematic diagram of the F-ASHP model. During the peak period of solar radiation P1 (12:00–16:00), the required temperature inside the pig houses for conservation increased, and the solar peak heat was stored in the energy storage pen. During the solar intermittent period P2 (16:00 to 12:00, the next day), the energy storage pen released heat, thereby reducing the heat load demand in the pig houses for conservation, transferring solar energy on an hourly scale, and achieving the goal of a flexible heat load demand in pig houses for conservation.

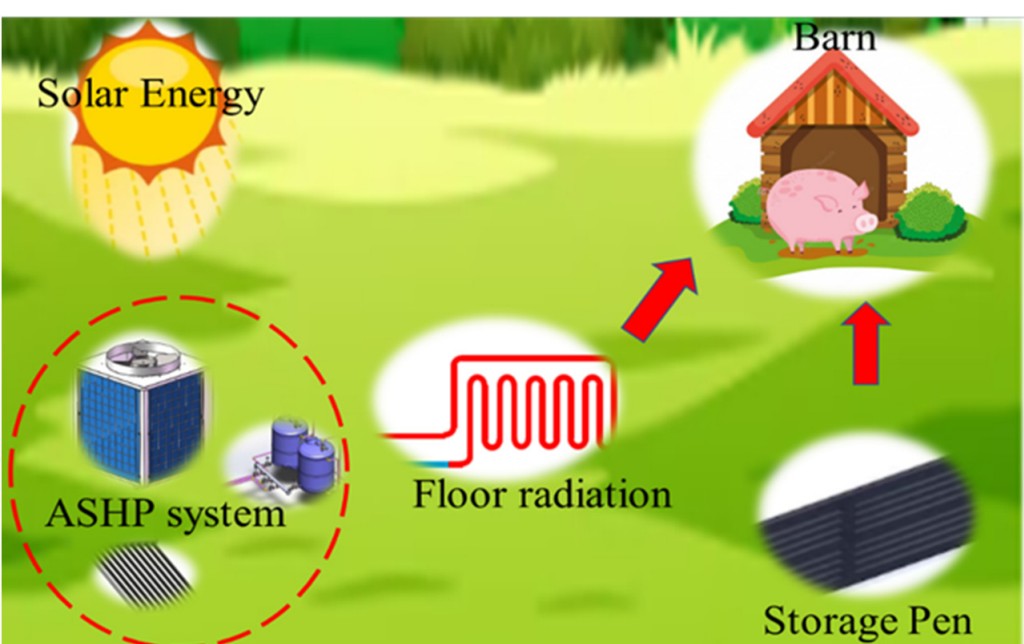

**Figure 1.** F-ASHP model.

This study constructs a mathematical model of the F-ASHP by coupling the ASHP, energy-storage pen, and solar collectors. The dynamic changes in heat storage, release time, and release amount in energy storage pens composed of different materials, as well as the dynamic change of heat load in pig houses for conservation, were studied with on-site testing and Simulink. Our findings provide the theoretical and technical foundation for reducing energy consumption and carbon emissions during the winter.

## 2. Materials and Methods

### 2.1. Geometric Model

The ASHP heating system was installed in a nursery pig house in Shunyi District, Beijing, China. The experimental pig house has a north–south orientation, and its peripheral dimensions are 42 m long and 9.3 m wide, as shown in Figure 2. There are 14 pens (3 m width and 7.5 m length each) in the pig house, with a 2 m × 3 m slatted floor area in each pen. The slat is 80 mm wide, and the opening is 20 mm wide. The pig house has two units, and each unit holds seven pens. The pig house is naturally ventilated. The wall of the pig house is made up of 0.37 m thick bricks and 90 mm thick extruded polystyrene board (for external insulation). The eave of the pig house is 2.9 m high, and the double pitch roof's material is color steel laminboard with a thickness of 100 mm. The windows are made of plastic and steel with two glass panes. There are 13 windows (2.00 × 1.45 m²) and 1 window (1.15 × 1.15 m²) in the southern wall, and 14 windows (1.00 × 0.80 m²) in the northern wall. There is one window with a size of 1.18 × 1.07 m² and one window with a size of 0.74 × 1.17 m² in the eastern wall. There is one wooden door (0.90 × 1.77 m²) in the western wall. There were 442 piglets in the two units of the pig house, and the age of the piglets was 61 d. The physical parameters of the building are shown in Table 1.

### 2.2. ASHP System

The thermal load of the pig house was 12.0 kW when the outdoor design temperature for heating was −7.6 °C in Beijing. Considering the efficiency of pipeline transportation and the safety factor of 1.3, the thermal load of the pig house needs to be configured at 15.6 kW. For heating the pig house, experimental heating equipment based on enhanced vapor injection technology, a type of low-temperature ASHP, was utilized. The outdoor extreme minimum temperature in Beijing can fall below −15 °C; thus, the heating capacity of the ASHP was 22.4 kW when the dry ball temperature was −7 °C and 20.2 kW when

the dry ball temperature was −15 °C (parameters from the equipment factory). The ASHP utilized R410A refrigerant and was installed in a water tank with a thermal storage capacity of approximately 1000 L. The hot water stored in the tank was pumped to heat the pig house. In the event that the water temperature in the tank dropped below the set temperature, the ASHP would heat the water in the circulating system to supply hot water to the tank. When the water temperature in the tank was higher than the setting temperature, the ASHP would stop working, but the water in the tank would still be pumped to heat the pig house. The ASHP heating system is shown in Figure 3.

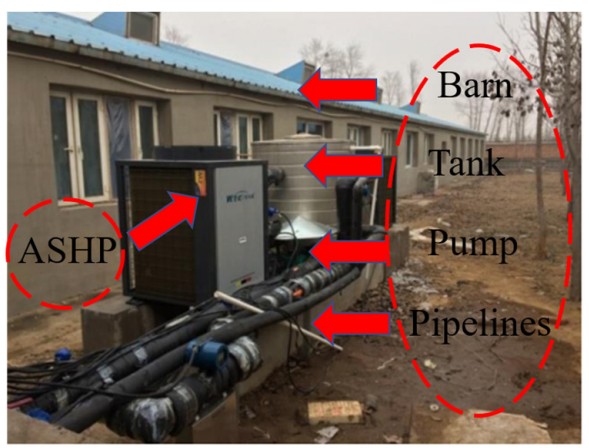 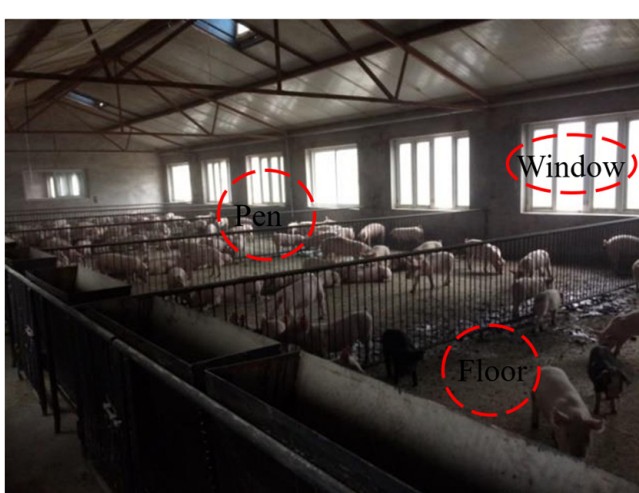

**Figure 2.** Experimental pig barn.

**Table 1.** Structure information.

| Structure | Direction | Area (m²) | Heat Transfer Coefficient (W/m²·K) |
|-----------|-----------|-----------|-------------------------------------|
| Door | West | 1.59 | 4.7 |
| Window | East | 1.26 | 1.92 |
| | South | 39.02 | 4.7 |
| | West | 0.87 | 4.7 |
| | North | 11.2 | 4.7 |
| Wall | East | 36.52 | 0.35 |
| | South | 83.79 | 0.35 |
| | West | 35.32 | 0.35 |
| | North | 11.62 | 0.35 |
| Floor | / | 396.52 | 0.22 |
| Roof | / | 380.52 | 0.47 |

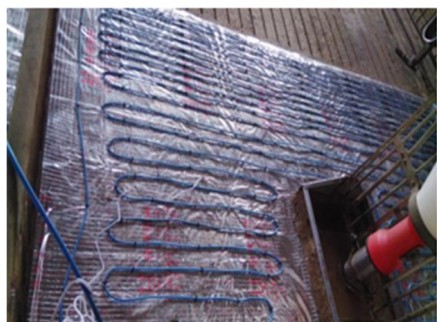

(**a**) Radiant floor          (**b**) ASHP

**Figure 3.** ASHP systems.

### 2.3. Mathematical Models

The assumption was based on the heat losses in the ASHP system from the piping and heating water tank, and this study was based on a certain stocking density for 20 kg pigs. Figure 4 is a schematic diagram of the heat balance equations for the F-ASHP model, including the heat supply of the ASHP, heat supply of solar collectors, heat loss of the water tank, heat transfer through the enclosure structure of pig houses, pig heat production, ventilation heat loss, and heat release from the energy storage pen. Equations (1)–(7) represent the heat balance equations for each component of the F-ASHP model. A flexible dynamic simulation model of the F-ASHP was built based on MATLAB/Simulink (Figure 5), and the relative tolerance was less than $10^{-6}$. Reference [12] describes the changes in the outdoor temperature and COP during the test period of the ASHP. During the test period, the heat storage water tank temperatures of 40 °C and 38 °C were used as the starting and stopping temperatures for the air-source heat pump, respectively.

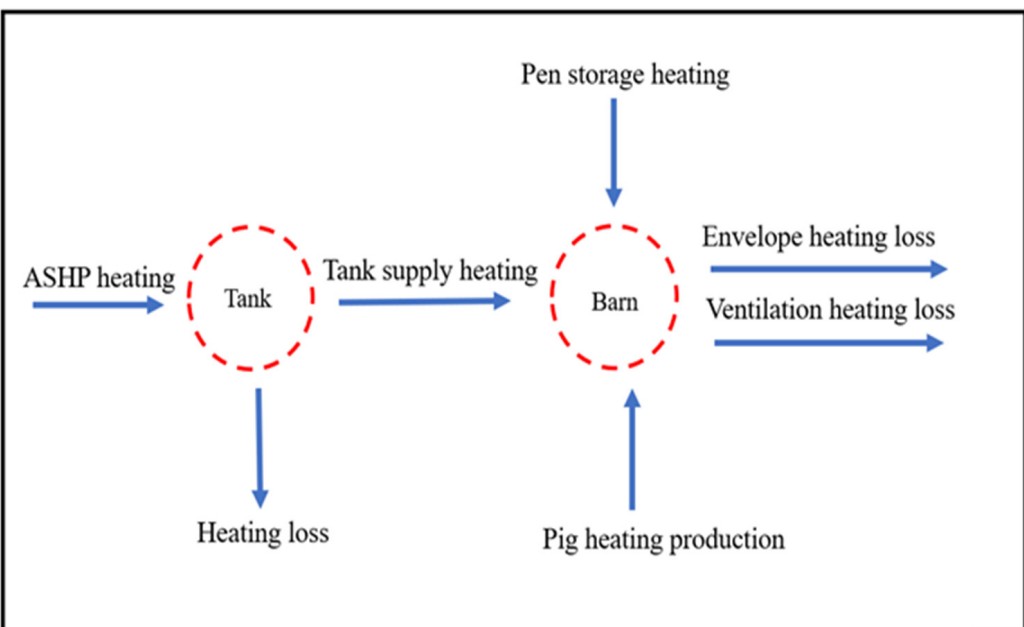

**Figure 4.** Heat balance schematic.

The major equations that were implemented in the subsystem of the Simulink model for F-ASHP were Equations (1) and (7), respectively [26].

$$C_p \times M_{TANK} \times \frac{dt}{d\tau} = Q_s + Q_{ASHP} - G \times C_p \times (T_{out} - T_{in}) - Q_{Loss} \tag{1}$$

where $C_p$ is the water of specific heat capacity, J/kg·K; $M_{TANK}$ is the quality of water tank, kg; $Q_s$ is the solar energy, W; $Q_{ASHP}$ is the heat of the ASHP, W; $G$ is the quality flow rate, kg/s; $T_{out}$ is the supply temperature of the tank, °C; $T_{in}$ is the return temperature of the tank, °C; and $Q_{LOSS}$ is the heat loss of the tank, W.

$$G \times C_p \times (T_{out} - T_{in}) = K_i \times A_b \times T_{barn} - T_{out} + Q_{ventilation} - Q_{pig} + Q_{source} \tag{2}$$

where $G$ is the quality flow rate, kg/s; $C_p$ is the water of specific heat capacity, J/kg·K; $T_{out}$ is the supply temperature of the tank, °C; $T_{in}$ is the return temperature of the tank, °C; $K_i$ is the heat transfer coefficient of enclosure, W/m²·K; $A_b$ is the area of the barn, m²; $Q_{ventilation}$ is ventilation heat loss, W; $Q_{pig}$ is the pig heat production, W; and $Q_{source}$ is the energy storage pen of heat, W;

$$T_a = T_{a,p} + \left(T_{a,min} - T_{a,p}\right) \times \cos\left[\frac{2\pi}{N}(\tau - 3)\right] \tag{3}$$

where $T_a$ is the ambient temperature, °C; $T_{a,min}$ is the min ambient temperature, °C; $T_{a,p}$; is the average ambient temperature, °C; and $N$ is time, h.

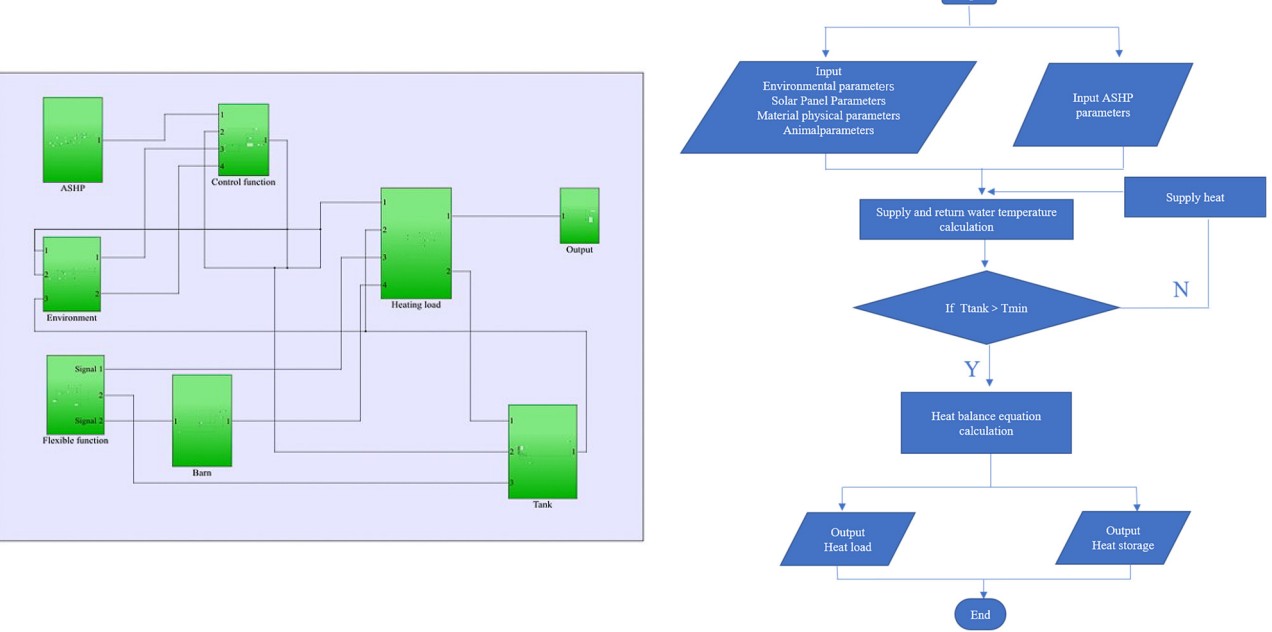

**Figure 5.** F-ASHP of the Simulink model.

$$\frac{dT_b}{d\tau} = F_R \times \left[(S - U_L) \times \left(T_{f,i} - T_a\right)\right] + G \times C_p \times \left(T_{f,i} - T_{f,o}\right) \tag{4}$$

where $T_{f,i}$ is the heat collector inlet temperature, °C; $T_{f,o}$ is the heat collector outlet temperature, °C; $T_a$ is the average ambient temperature, °C; $U_L$ is the heating loss, W/m²·°C; $S$ is the solar radiation intensity, W/m²; and $F_R$ is the heat transfer factor.

$$Q_{source} = C_{sp} \times M_S \times \frac{dt}{d\tau} = K \times A_p \times (T_s - T_{barn}) \tag{5}$$

where $C_{sp}$ is the energy storage pen of Specific heat capacity, J/kg·K; $M_s$ is the quality of energy storage pen, kg; $A_b$ is the Area of pen, m²; $T_s$ is the average temperature of energy storage pens, °C; and $K$ is the heat transfer coefficient of energy storage pen, W/m²·K.

$$Q_{pig} = \frac{1000 + 12 \times (20 - T_a)}{1000}\left\{5.09 \times m^{0.75} + [1 - (0.47 + 0.003 \times m)] \times \left[n \times 5.09m^{0.66} - 5.09m^{0.66}\right]\right\} \tag{6}$$

where $m$ is the quality of nursery pig, kg; $n$ is the ratio of energy obtained to the maintenance energy of the nursery pig; and $T_a$ represents air temperature, °C.

$$Q_{ventilation} = C_{pa} \times V \times \rho \times T_{barn} - T_{out} \tag{7}$$

where $C_{pa}$ is the air of specific heat capacity, J/kg·K; $\rho$ is air density, kg/m³; and $V$ is the ventilation rate of the nursery pig, m³/h.

### 2.4. Boundary Conditions

This study proposed an F-ASHP model in a nursery pig house, and 15 cases were used to simulate the heat storage and release, heat storage and release times, and variation in pig house heat load for different energy storage pen materials under 40 °C water tank set

temperatures. The four storage pen materials are concrete, gray sand brick, magnesium oxide heat storage brick, and clay. The pen dimensions are 4.8 m in length, 2.4 m in width, and 0.03 m in thickness. The storage pen's materials and simulation conditions are listed in Tables 2–4.

**Table 2.** Storage pen materials.

| Materials | Density (kg/m$^3$) | Specific Heat Capacity (J/kg·K) | Heat Exchange Area (m$^2$) | Quality (kg) |
|---|---|---|---|---|
| Concrete brick | 2300 | 920 | 1.92 | 132.48 |
| Gray sand brick | 1900 | 1050 | 1.92 | 109.44 |
| Clay brick | 1842 | 1850 | 1.92 | 106.10 |
| Magnesium oxide Storage brick | 2500 | 1140 | 1.92 | 144.00 |

**Table 3.** Model parameters.

| Parameter | Value | Parameter | Value |
|---|---|---|---|
| Water tank volume | 1000 L | Number of pigs | 448 |
| Water specific heat capacity | 4.2 kJ/kg·°C | Ventilation rate | 5.8 m$^3$/h |
| Air specific heat capacity | 1.005 kJ/kg·k | Collector area | 6 m$^2$ |
| Air density | 1.29 kg/m$^3$ | Energy utilization rate | 0.8 |
| Water density | 1000 kg/m$^3$ | Heat transfer factor | 0.9 |
| Nursery pig weight | 13.6 kg | Barn set temperature | 25 |
| Average daily solar radiation | 9.85 MJ/d | Average outdoor temperature | −7.9 |
| Lowest temperature | −18.3 °C | Flow rate | 110 L/h |

**Table 4.** Simulation conditions.

| Case | Tank Setting Temperature (°C) | Pen Material | Solar Energy | P1 Setting Temperature (°C) |
|---|---|---|---|---|
| 1 | 40 | Hollow plastic | / | 26 |
| 2 | | | | 27 |
| 3 | | | | 28 |
| 4 | | Clay brick | Working | 26 |
| 5 | | | | 27 |
| 6 | | | | 28 |
| 7 | | Gray sand brick | Working | 26 |
| 8 | | | | 27 |
| 9 | | | | 28 |
| 10 | | Magnesium oxide Storage brick | Working | 26 |
| 11 | | | | 27 |
| 12 | | | | 28 |
| 13 | | Concrete brick | Working | 26 |
| 14 | | | | 27 |
| 15 | | | | 28 |

*2.5. ASHP System Measuring Points*

The ASHP heating system and indoor environment were monitored during the period between 15 November 2016 and 16 April 2017. The water temperatures were monitored by Pt100 temperature sensors (KZW/P-231, measuring range: 0~70 °C, accuracy: ±0.15 °C, KunLunZhongDa Sensors Co., Ltd., Beijing., China). All the data on water temperature and flow were displayed and recorded by a PLC system.

Air temperature and RH indoors and outdoors were monitored by a temperature and humidity automatic recorder (Apresys, 179A-TH, measuring range, RH: 0%~100%, temperature: −40~100 °C, accuracy: ±2% and ±0.2 °C, San Ramon, CA, USA). Ground tem-

perature was monitored by ground thermometers (JTR09, measuring range: −20~85 °C, accuracy: <±4%, Tian Jian Hua Yi Co., Ltd., Beijing., China).

## 3. Results and Discussion

### 3.1. Model Validation

We referenced [11], monitored, and analyzed the ASHP supply and return water temperatures and the indoor environment of pig houses. A higher relative humidity will raise the dew-point temperature of the air, and air condensation will frost on the surface of the ASHP evaporator, affecting heat exchange. The lowest temperature in the Beijing area in the last 30 years was −11.8 °C. A defrosting device was activated after the evaporator surface reached the dew-point temperature, and condensation could occur in colder climate zones, which would affect the ASHP system.

To verify the accuracy of the simulation model, we selected Case 1. Figure 6 illustrates the measured and simulated values of the ASHP system's supply and return water temperatures on 4 January. The simulation results for the supply and return water temperatures of the ASHP heating system in the pig houses exhibited a similar changing trend to the measured values. During the noon period, the supply and return water temperatures were lower than those in other time periods. These values were attributed to the higher outdoor temperature during noon, which reduced the heat transfer from the pig houses to the outdoors and consequently decreased the heat load required by the pig houses. The maximum relative error between the simulated and measured values of the supply and return water temperatures was 5.1%. This fluctuation was attributed to short-term fluctuations in the outdoor temperature between 15:00 and 16:00, which deviated from the change trajectory of the outdoor temperature wave function. Overall, this mathematical model can be effectively utilized for numerical heat transfer research in pig houses for conservation purposes.

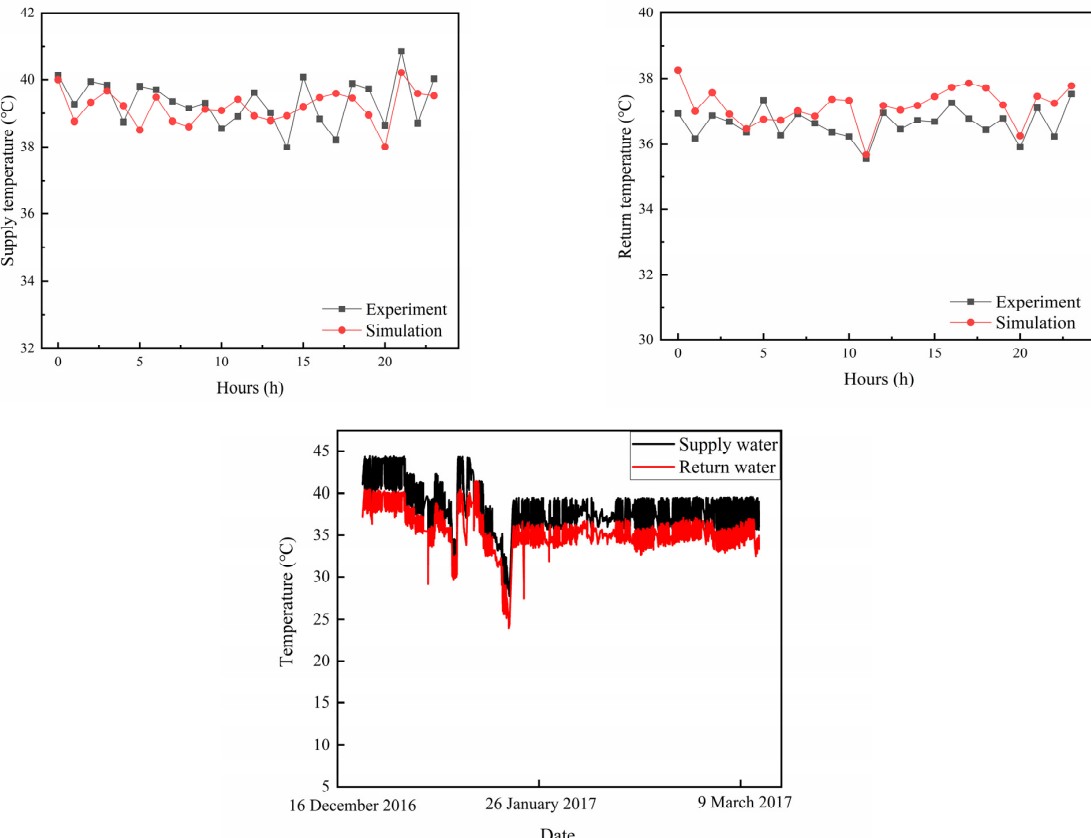

**Figure 6.** Experimental data and simulation validation of the model.

### 3.2. Analysis of Heat Storage and Release of the Pen

#### 3.2.1. Analysis of Heat Storage

Cases 5, 8, 11, and 14 were selected to analyze the heat storage in the F-ASHP model. Figure 7 shows the heat storage of the four types of energy storage pens in operating hours, which demonstrated the same trend. The heat storage time required for the brick pen to increase by 2 °C was 14.4 h, whereas the heat storage time for the clay pen to rise by 2 °C was 15.0 h. The temperature change rate of the gray sand brick storage pen was the fastest, owing to its smallest specific heat capacity. The temperature rise rate of the storage pen of the four materials increased rapidly in the first 3.5 h, which could be explained by the larger initial temperature difference leading to a larger heat transfer rate. At 4 h of heat storage time, the average temperatures of the four types of material pens were 26.94 °C, 26.92 °C, 26.73 °C, and 26.82 °C, respectively.

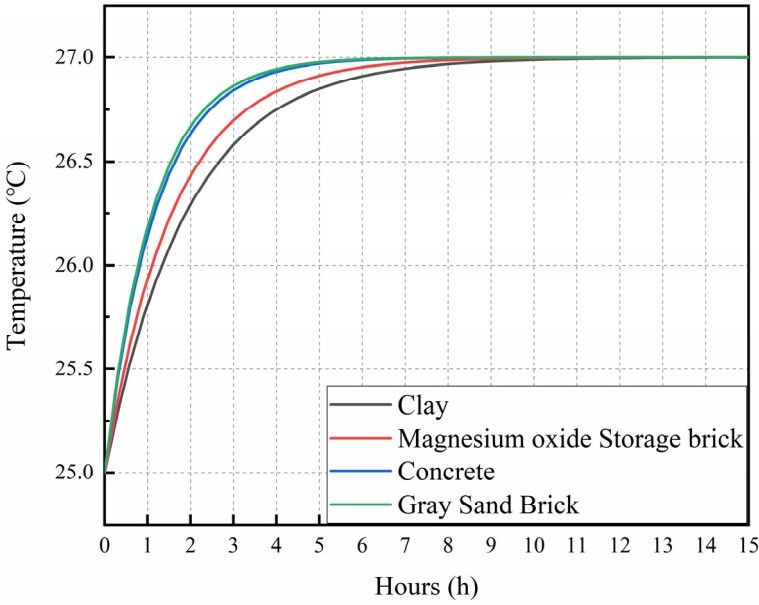

**Figure 7.** The heat storage of four pen materials.

#### 3.2.2. Analysis of Heat Release

Figure 8 shows the heat release of the energy storage pens composed of four different materials (Cases 5, 8, 11, and 14) during the solar intermittent period. In the first 30 min of heat release, the gray sand brick storage pen released the most heat. As the heat release time increased, the heat release of the clay pen was higher than those of the other three materials, which was attributed to the highest specific heat capacity of the clay energy storage pig pen. During the solar intermittent period, the total heat release of individual clay energy storage pig pens, magnesium oxide storage brick pens, concrete brick energy storage pens, and gray sand brick energy storage pens was 374.44 kJ, 331.92 kJ, 270.72 kJ, and 260.64 kJ, respectively. This phenomenon indicates that the energy storage pen can effectively store energy during the solar peak period and achieve energy transfer on an hourly scale.

However, while the clay brick energy storage pig pen offered the best heat release effect, it can be adversely impacted by certain factors, such as cleaning and disinfection in the pig nursery. Therefore, we decided to use the magnesium oxide storage brick pen in the pig nursery.

Previous studies [27] have shown that pen attachment ventilation (PAV) for convective heating has better effects than ASHP floor radiant heating. PAV can directly perform convective heat exchange with the energy storage pen, and the combination of PAV and the energy storage pen may be increasingly conducive to energy storage during the solar peak period.

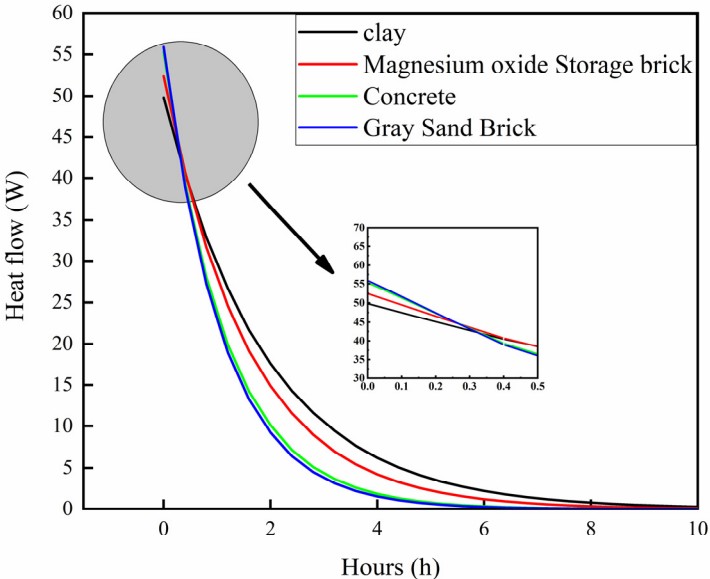

**Figure 8.** The heat release of four pen materials.

### *3.3. The Analysis of F-ASHP Heat Load*

Figure 9 illustrates the heat load of the nursery pig house for Cases 1, 8, and 11. In Case 1, no flexible heat load regulation was observed, and the barn setting temperature was maintained at 25 °C. In Cases 8 and 11, the barn setting temperature increased during the P1 stage. The heating load in all three cases decreased during the P1 stage due to the rising outdoor temperature, which reduced the heat transfer within the enclosure. Cases 8 and 11 had higher heat loads during the P1 stage compared with that in Case 1 because they utilized solar energy for pig pen heat storage. During the P2 stage, the peak-valley heat loads for Cases 1, 8, and 11 were 7.80 kW, 7.31 kW, and 7.01 kW, respectively. Both Case 2 and Case 11 had lower heat loads than Case 1, indicating that the energy storage pen effectively facilitated hourly energy transfer, reducing the indoor heat load demand during periods of solar intermittency. The heat loads of Cases 8 and 11 in the P2 stage exhibited a similar trend. However, Case 8 had a higher heat load during the P2 stage compared with that in Case 11, attributable to the lower heat capacity of the energy storage pen in Case 8, which stored less heat during the P1 stage.

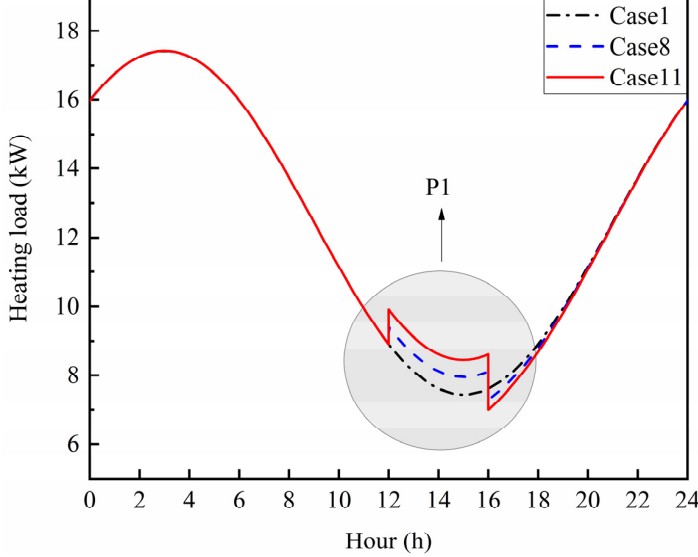

**Figure 9.** The heat load curve under different working conditions.

The decrease in barn heat load was highly pronounced during the early stages of P2, as the temperature of the energy storage pig pen was higher than the indoor temperature, resulting in a faster heat transfer rate. Concurrently, the outdoor temperature gradually decreased with the setting sun, increasing the heat transfer within the indoor enclosure. Compared with that in Case 1, the use of a magnesium oxide energy storage pen with a 2-degree increase during the P1 stage reduced the heat load of the ASHP by 10.1% in the P2 stage, highlighting the elasticity in heat load demand achieved by the F-ASHP model in the pig nursery. However, not all the heat added in the P1 stage was transferred to the P2 stage, as some of it was lost through the enclosure, and the insulation effect of the enclosure affected the energy storage during the P1 stage. In addition, the energy storage pen and F-ASHP were suitable for regions with abundant sunlight, whereas cloudy conditions could affect the energy storage and transfer efficiency of the energy storage pig pen.

## 4. Conclusions

In this study, a flexible heating model combining an ASHP and energy storage pig pens was proposed. Using a nursery pig house in Beijing, simulations were conducted to analyze different material energy storage pens, heat storage capacity, heat storage time, and variations in barn heat load using on-site testing and Simulink. The main conclusions are summarized as follows:

(1) The F-ASHP model for a nursery pig house can accurately predict the heat load demand with a relative error of 5.1%. This model can be used to evaluate the feasibility of F-ASHP in different regions with abundant solar energy.

(2) The energy storage pen effectively stores heat during solar peak times and releases it during solar intermittency, achieving hour-scale energy transfer. Compared with the ASHP system, the magnesium oxide energy storage pen reduced the heat load demand in the pig house by 10.1%, resulting in a highly flexible barn heat load demand.

(3) Among the four energy storage pens, the magnesium oxide storage brick demonstrated the best performance, with a total accumulated heat storage of 3319.20 kJ.

## 5. Limitations

This study proposed that the F-ASHP heating model could achieve hour-scale energy transfer of solar energy and reduce the internal heat load of nursery pig houses during solar intermittency through the use of an energy storage pen. However, this study still has some limitations. Principally, it investigated the F-ASHP model using theoretical calculations, where differences in its actual application in pig production may be present. Additionally, energy storage pens are made of solid materials, and their heat storage capacity is affected by the specific heat capacity of the specific materials used. Therefore, future studies should further monitor and analyze the indoor thermal environment of the F-ASHP heating system and investigate a flexible heating model of phase-change energy storage for pig pens coupled with a PAV.

**Author Contributions:** Data curation, L.Y., J.L. (Jia Liu) and S.L.; investigation, H.W.; methodology, H.W., J.L. (Jijun Liu) and Z.W.; project ad-ministration, M.W.; software, H.W.; supervision, M.W.; writing—original draft, H.W.; writing—review and editing, H.W., Y.L., S.L. and M.W. All authors have read and agreed to the published version of the manuscript.

**Funding:** This research is supported by the Science and Technology Innovation 2023—"New Generation Artificial Intelligence" Major Project of China (2022ZD0115704).

**Institutional Review Board Statement:** Not applicable.

**Data Availability Statement:** All datasets from the current study are available from the corresponding author upon reasonable request.

**Conflicts of Interest:** The authors declare that they have no known competing financial interests or personal relationships that could have appeared to influence the work reported in this paper.

## Nomenclature

*Symbols*

| | |
|---|---|
| $A_b$ | Area of barn, m$^2$ |
| $A_p$ | Area of pen, m$^2$ |
| $C_{pa}$ | Air of specific heat capacity, J/kg·K |
| $C_p$ | Water of specific heat capacity, J/kg·K; |
| $C_{sp}$ | Energy storage pen of specific heat capacity, J/kg·K |
| $F_R$ | Heat transfer factor |
| $G$ | Quality flow rate, kg/s |
| $K$ | Heat transfer coefficient of energy storage pen, W/m$^2$·K |
| $K_i$ | Heat transfer coefficient of enclosure, W/m$^2$·K |
| $m$ | Quality of nursery pig, kg |
| $M_s$ | Quality of energy storage pen, kg |
| $M_{TANK}$ | Quality of water tank, kg |
| $Q_{ASHP}$ | Heat of the ASHP, W |
| $Q_{Loss}$ | Heat loss of the tank, W |
| $Q_{pig}$ | Pig heat production, W |
| $Q_s$ | Solar energy, W |
| $Q_{source}$ | Energy storage pen of heat, W |
| $Q_{ventilation}$ | Heat loss of Ventilation, W |
| $S$ | Solar radiation intensity, W/m$^2$ |
| $T_a$ | Ambient temperature, °C |
| $T_{a,p}$ | Average ambient temperature, °C |
| $T_{a,min}$ | Min ambient temperature °C |
| $T_{barn}$ | Supply air temperature, °C |
| $T_{f,i}$ | Heat collector inlet temperature, °C |
| $T_{f,o}$ | Heat collector outlet temperature, °C |
| $T_{in}$ | Return temperature of the tank, °C |
| $T_{out}$ | Supply temperature of the tank, °C |
| $T_s$ | Average temperature of energy storage pens, °C $^2$ |
| $U_L$ | Heating loss, W/m$^2$·°C |
| $V$ | Ventilation rate of nursery pig, m$^3$/h. |
| $\rho$ | Air density, kg m$^{-3}$ |

*Abbreviations*

| | |
|---|---|
| ASHP | Air-source heat pump |
| F-ASHP | Flexible air-source heat pump system |
| COP | Coefficient of performance |
| PAV | Pen-attached ventilation |

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
