# Peer review of "Research on the Flexible Heating Model of an Air-Source Heat Pump System in Nursery Pig Houses"

_agriculture, doi:10.3390/agriculture13051059_

Round 1

Reviewer 1 Report

My comments are:

1. The Abstract must contain the most important numerical values obtained in this study.

2. English language should be revised in many places within the manuscript.

3. Table 1: the unit of heat transfer coefficient is W/m2K not w/m2K.

4. After equations1, 2, 5 and 7: j should be J in the specific heat unit and Specific should be specific.

5. Define K after equation 5.

6. Why the pen attachment ventilation (PAV) for convective heating have better effects compared to ASHP floor radiant heating.

7. What are the thermal conductivities of the materials used in this study?

8. What is the effect of outdoor air humidity on the system performance?

9. The literature review should be improved to show the importance of heat pumps and their applications. Such information is available from

https://doi.org/10.1016/B978-0-12-821602-6.00013-4

There are many typo errors regarding the units.

Author Response

Manuscript Response Letter

To

Agriculture

Manuscript Title: Research on the flexible heating model of air source heat pump system in nursery pig house

Dear Editors and Reviewers:

Thanks for your letter and for reviewers’ comments concerning our manuscript entitled " Research on the flexible heating model of air source heat pump system in nursery pig house" (Manuscript ID: agriculture-2390095). Those comments are all valuable and helpful for revising and improving our paper. We have studied all comments carefully and have made conscientious correction. Revised portion are marked in red in the paper. The main corrections in the paper and the response to the reviewers’ comments are as flowing:

Comments and Suggestions for Authors

My comments are:

  1. The Abstract must contain the most important numerical values obtained in this study.

Response: We are so grateful for your kind question. It is our negligence and we are sorry about this. According to comment, related content has been improved. We have re-written this part according to the Reviewer’s suggestion. I hope you will agree. (Line7-20)

  1. English language should be revised in many places within the manuscript.

Response: We regret there were problems with English. The paper has been carefully revised by a professional language editing service to improve the grammar and readability.

  1. Table 1: the unit of heat transfer coefficient is W/m2K not w/m2K.

Response: It is our negligence and we are sorry about this. All the modifications are marked as red. We hope that you agree. (Line128 Table.1)

  1. After equations1, 2, 5 and 7: j should be J in the specific heat unit and Specific should be specific.

Response: We sincerely appreciate the valuable comments, and we have made correction according to the Reviewer’s comments. (Line 166,171,184 and 193)

  1. Define K after equation 5.

Response: We were really sorry for our careless mistake. Thank for your reminder. (Line 187)

  1. Why the pen attachment ventilation (PAV) for convective heating have better effects compared to ASHP floor radiant heating.

Response: PAV is a convection heating technology that directly regulates the animal AOZ. Compared with radiant heating, firstly, it can weaken the heat load in the background space of the barn, and secondly, it reduces the influence of thermal inertia of the barn envelope and does not require additional fresh air system to regulate the air quality of the barn. Therefore, we think that PAV is better than radiant heating in terms of heating effect. I hope you can agree with my point of view.

  1. What are the thermal conductivities of the materials used in this study?

Response: We apologize for the confusion caused by the conversion of the table format. We have made the correction and apologize again.(Table 1-2)

  1. What is the effect of outdoor air humidity on the system performance?

Response: We are so grateful for your kind question. A higher relative humidity will raise the dew-point temperature of the air, and air condensation will frost on the surface of the ASHP evaporator, affecting heat exchange. The lowest temperature in Beijing area in the last 30 years was -11.8℃. A defrosting device was activated after the evaporator surface reached the dew-point temperature, and condensation could occur in colder climate zones, which would affect the ASHP system. (Line226-231)

  1. The literature review should be improved to show the importance of heat pumps and their applications. Such information is available from

https://doi.org/10.1016/B978-0-12-821602-6.00013-4

Response: We thank the reviewers for their valuable suggestions, and these references are of great research importance. We have made correction according to the Reviewer’s comments. (Line 81)

Comments on the Quality of English Language

There are many typo errors regarding the units.

Submission Date

24 April 2023

Date of this review

30 Apr 2023 11:19:04

Special thanks to you for your good comments. We tried our best to improve the manuscript and made some changes in the manuscript. These changes will not influence the content and framework of the paper. We appreciate for Editors/Reviewer’ warm work and hope that the correction will meet with approval.

Once again thank you very much for your comments and suggestions.

                                                                                                             Hua Wang

Reviewer 2 Report

Line number must be included

Pg. 2 line 20 "Park [21] conducted experimental studies on the thermal performance of phase-change bricks doped with phase-change materials in summer and winter environments" this sentence has some error modify it

Figure 2 must be modified by including marking of components/parts

Check the unit of heat transfer coefficient in Table 1, the values are thermal conductivity or convective heat transfer coeff ? Give proper reference

Give details about refrigerant used in the Heat Pump and heat pump specifications

Give separate nomenclature at the beginning of the article

Give reference for equation 6

List the assumptions for modelling ASHP

Give a flowchart representing solution procedure for the equations

Give proper captions for figures

What does hour mentioned in x-axis of graphs 7,8,9 represents? Whether iit is operating hour or solar hour

Experimental results must be plotted

English language usage must be improved

Author Response

                                              Manuscript Response Letter

                                                           To

                                                   Agriculture

Manuscript Title: Research on the flexible heating model of air source heat pump system in nursery pig house

Dear Editors and Reviewers:

Thanks for your letter and for reviewers’ comments concerning our manuscript entitled " Research on the flexible heating model of air source heat pump system in nursery pig house" (Manuscript ID: agriculture-2390095). Those comments are all valuable and helpful for revising and improving our paper. We have studied all comments carefully and have made conscientious correction. Revised portion are marked in red in the paper. The main corrections in the paper and the response to the reviewers’ comments are as flowing:

1. 2 line 20 "Park [21] conducted experimental studies on the thermal performance of phase-change bricks doped with phase-change materials in summer and winter environments" this sentence has some error modify it

Response: We regret there were problems with English. The paper has been carefully revised by a professional language editing service to improve the grammar and readability. Line 71-74, the statement of "Park [21] conducted experimental studies on the thermal performance of phase-change bricks doped with phase-change materials in summer and winter environments." were corrected as " Furthermore, Park [21] conducted experimental investigations on the thermal performance of phase-change bricks doped with phase-change materials in both summer and winter environments, comparing their effectiveness in controlling indoor temperatures with different roof materials."

2.Figure 2 must be modified by including marking of components/parts

Response: we appreciate the reviewer for this kind recommend. It is our negligence and we are sorry about this. According to comment, related content has been improved. Missing parts have been added.

3.Check the unit of heat transfer coefficient in Table 1, the values are thermal conductivity or convective heat transfer coeff ? Give proper reference

Response: It is our negligence and we are sorry about this. and we have made correction according to the Reviewer’s comments. All the modifications are marked as red. We hope that you agree. (Line128 Table.1)

4.Give details about refrigerant used in the Heat Pump and heat pump specifications

Response: We sincerely appreciate the valuable comments, and we have made correction according to the Reviewer’s comments. Line 137-138, " The ASHP utilized R410A refrigerant and was installed in a water tank with a thermal storage capacity of approximately 1000 L. " was added. We hope that you agree.

5.Give separate nomenclature at the beginning of the article

Response: We sincerely appreciate the valuable comments, and we have made correction according to the Reviewer’s comments. The nomenclature was added.

6.Give reference for equation 6

Response: We sincerely appreciate the valuable comments, and we have made correction according to the Reviewer’s comments. The reference was added. (Line 165)

7.List the assumptions for modelling ASHP

Response: We deeply agree with you and have incorporated this suggestion throughout our paper. Line 148-150, " The assumption was based on the heat losses in the ASHP system from the piping and heating water tank, and this study was based on a certain stocking density for 20 kg pigs. " was added. We hope that you agree.

8.Give a flowchart representing solution procedure for the equations

Response: We thank the reviewers for their valuable suggestions. The flowchart was added. We hope that you agree.

9.Give proper captions for figures

Response: We are so grateful for your kind question. It is our negligence and we are sorry about this. According to comment, related content has been improved.

10.What does hour mentioned in x-axis of graphs 7,8,9 represents? Whether it is operating hour or solar hour

Response: We sincerely appreciate the valuable comments. I apologize for not being able to express the meaning of x-axis of graphs, and it is operating hour.

11.Experimental results must be plotted

Response: We are so grateful for your kind question. It is our negligence and we are sorry about this. According to comment, related content has been improved. The experimental data are plotted in Figure 6.

Special thanks to you for your good comments. We tried our best to improve the manuscript and made some changes in the manuscript. These changes will not influence the content and framework of the paper. We appreciate for Editors/Reviewer’ warm work and hope that the correction will meet with approval.

Once again thank you very much for your comments and suggestions.

Round 2

Reviewer 2 Report

comments have been addressed